# COVID-19 and Lockdown, as Lived and Felt by University Students

**DOI:** 10.3390/ijerph192013454

**Published:** 2022-10-18

**Authors:** Cátia Branquinho, Fábio Botelho Guedes, Ana Cerqueira, Alexandra Marques-Pinto, Amélia Branco, Cecília Galvão, Joana Sousa, Luís F. Goulão, Maria Rosário Bronze, Wanda Viegas, Margarida Gaspar de Matos

**Affiliations:** 1Instituto de Saúde Ambiental (ISAMB), Faculdade de Medicina, Universidade de Lisboa (FMUL), 1649-028 Lisbon, Portugal; 2Equipa Aventura Social, 1400-185 Lisbon, Portugal; 3Faculdade de Motricidade Humana, Universidade de Lisboa (FMHUL), 1495-751 Lisbon, Portugal; 4Centro de Investigação em Ciência Psicológica, Faculdade de Psicologia, Universidade de Lisboa, 1649-013 Lisbon, Portugal; 5GHES Research Center—Office of Economic and Social History, ISEG—Lisbon School of Economics & Management of the University of Lisbon, 1249-078 Lisbon, Portugal; 6Instituto de Educação, Universidade de Lisboa (IEUL), 1649-013 Lisbon, Portugal; 7Laboratório de Nutrição, Universidade de Lisboa, 1649-028 Lisbon, Portugal; 8Unidade de Investigação Linking Landscape, Environment, Agriculture and Food (LEAF), Instituto Superior de Agronomia, Universidade de Lisboa (ISAUL), 1349-017 Lisbon, Portugal; 9Faculdade de Farmácia, Universidade de Lisboa (FFUL), 1649-003 Lisbon, Portugal; 10Instituto de Biologia Experimental e Tecnológica (iBET), 2780-157 Lisbon, Portugal; 11APPSYci, ISPA—Instituto Universitário, 1149-041 Lisbon, Portugal

**Keywords:** Portugal, pandemic, COVID-19, first-year university students, mixed method

## Abstract

In the last 2 years, the COVID-19 pandemic has spread all over the world, forcing the closure of universities, among other unusual measures in recent history. (1) Background: This work is based on the study HOUSE-ULisbon, a survey carried out during the second confinement (March–May 2021) in Portugal with the collaboration of all the Faculties of the University of Lisbon (UL). The present work aims to explore gender differences in how first-year college students experienced and felt COVID-19 and the second confinement. (2) Methods: A questionnaire was carried out. In total, 976 university students (19.66 years (*SD* = 4.033); Min = 17 and Max = 65) from the first year of the UL were included, of which 69.5% (*n* = 678) were female, and 30.5% were male (*n* = 298). SPSS v. 26 was used for quantitative data and MAXQDA 2020 for qualitative data. (3) Results: Overall, students reported various symptoms of physical and mental discomfort (especially females). Statistically significant differences were found in the problems that could arise from the pandemic, such as the prevalence of higher anxiety and worries by females, and online gaming by males. In coping strategies, differences were found in leisure and family relationships, with greater difficulty on the female side. Social interaction was perceived as difficult or very difficult by both genders. As strategies for future pandemics, they highlighted a concerted effort between the government and media in the transmission of messages to the population, facilitating information, knowledge and adoption of protective behaviors. (4) Conclusions: These results are important data for activating or maintaining resources and services for first-year university students, who in some university institutions were supported during the pandemic by psychological, material (e.g., computers, internet), and financial support measures, which are now diminished or extinct. The impacts on their lives will certainly not be extinguished post-pandemic, and health, education, and public policy measures should be prioritized for this group. These results are important data for activating resources and services for students, informing health and education professionals, and supporting public policies.

## 1. Introduction

Declared a pandemic disease on 11 March 2020 [1], the SARS-CoV-2 coronavirus has forced the implementation of numerous measures to mitigate the virus spread [2]. In Portugal, the first emergency state was declared on 18 March 2020 [3], as a result of a calamity situation, prompting mandatory confinement of patients or those infected with SARS-CoV-2; isolation of risk groups; and restriction of circulation on public roads to other citizens (who moved to a telework regime, whenever their duties allowed). Students, who were unable to pursue face-to-face education with the closure of educational institutions, experimented with an online methodology adopted by numerous countries [4,5]. Its impacts and methodology continue to be the focus of studies [6].

Restricted to their homes, with mandatory curfews and closure of all non-essential services, and with few resources from the public health system (a global scenario) [7], the physical and psychological, social, family, and academic consequences rapidly emerged in students’ lives [8,9,10,11,12,13,14,15].

University students, who had maintained a full-time or part-time online learning experience for a long time, seem to present, along with the female gender, more negative effects arising from the pandemic, and a more pessimistic view [13,16]. In a study conducted in Portugal in the post-confinement period, this group also reported feeling that their integration into university life had been constrained [13]. Several studies support the negative impacts of the pandemic on university student’s mental health (e.g., [17,18,19]), mainly in the increase in depressive and anxious symptomatology [20,21,22], reinforcing authors such as Azmi et al. [23] the view that online education associated with possible economic difficulties, or faculty-related concerns may contribute to this situation [24]. At the professional level, Maqsood et al. also found a relationship between a higher level of perceived stress, reduced salary, and remote work [25].

A study focusing on the effects of the pandemic on young people’s lifestyles, found a decrease in physical activity, an increase in sedentary behaviors, and increased screen time [26]. These findings may be associated with the higher levels of psychological distress and stress experienced, which affect behavioral mechanisms, and may have led to changes in sleep patterns, eating, and practice of physical activity [27].

When facing changes in behavioral patterns and psychological distress, motivated by uncertainties, increased conflicts with family, isolation from friends, or increased learning workload, students have an urgent need to adopt positive coping strategies, such as acceptance, planning, or seeking emotional or psychological support [28], rather than negative ones, such as substance use (e.g., tobacco) [16] or excessive online gaming [29].

A previous study, also including Portuguese university students, during the first wave of the pandemic [9] identified five messages for future pandemics: (i) to follow the guidelines of the Directorate-General of Health; (ii) to improve the preparation of the National Health System to deal with pandemics; (iii) to better prepare teachers for online teaching, with appropriate methodologies; (iv) to stimulate the fulfillment of daily routines (e.g., nutrition and physical activity) with the support of television; (v) and gaining awareness of the value of freedom.

In an unprecedented scenario with still unpredictable medium and long-term effects, namely in the development process, this study aims to investigate gender differences in how first-year college students experienced and felt COVID-19 and the second confinement.

## 2. Materials and Methods

### 2.1. Design and Participants

In the framework of Project HOUSE-Colégio F3 [30] ULisboa, a study on the health and lifestyle profile of first-year students at the University of Lisbon, Portugal, was carried out between March and May 2021 (vaccination and phased de-confinement). Focusing on the characterization of the university population, in the dimensions of health and well-being, psychotropic and potentially addictive substances, physical activity, eating habits, physical and emotional issues, literacy and knowledge, the present study only includes the questions related to the pandemic COVID-19. The average response time to the full instrument was 30′–40′. The questionnaire was carried out online and anonymously, spread via email by students in each faculty.

A total of 976 participants (M = 19.66 years old ± 4.033; Min = 17 and Max = 65) from 17 faculties or institutes of the University of Lisbon were considered, where 89% had entered the 1st year in the academic year 2020/2021 and 10.9% in previous academic years. In addition, 69.5% were female; 93.3% were Portuguese and 6.7% from other nationalities; and 90.8% had exclusive student status (6.3% part-time and 3% full-time workers).

When questioned about the educational level of their parents, 42.2% of the fathers and 56.1% of the mothers had a university education, 30% of the fathers and 25.9% of the mothers had completed secondary education and 19.7% of the fathers and 13.4% of the mothers had completed the 2nd/3rd cycle of basic education.

This study was approved by the Ethics Committee of the Centro Académico de Medicina de Lisboa, Centro Hospitalar de Lisboa Norte, EPE.

### 2.2. Measures

An online survey conducted through the Google Forms platform was designed based on the following questionnaires: Nutrition [31], Health Behaviour in School-aged Children/World Health Organization [32,33], and Health Literacy [34].

The questions considered in the present work regarded the university students’ perspectives about (1) how COVID-19 had affected their well-being (open-ended question); (2) what problems had come up, in areas such as sedentary behavior, poor nutrition, sleep quantity, sleep quality, substance use, depression, anxiety, worry, online gaming, excessive screen use, family conflicts (Yes or No questions), and other (open-ended question); (3) what strategies had helped them to cope with the COVID-19, whether the family bonds, having more time, feeling less fatigue (Yes or No) or other (open-ended question); (4) how easy/difficult was it to cope with daily routines and to balance work/study and leisure time, family interactions, socializing (Likert scale with 5 points where 1 = very easy and 5 = very difficult); (5) what were the relevant messages for future pandemics (open-ended question).

### 2.3. Data Analysis

A mixed methodology was used in this study, based on the nature of the data. Quantitative data were transported to SPSS v. 26 software and analyzed using the Chi-square test. A significance value of *p* < 0.05 was established.

The qualitative data, after a first content analysis, were studied using MAXQDA 2020, through the rule of thumb principle (line-by-line procedure). The thematic categories created had their origin in the impacts on well-being (Category 1), the problems that appeared (Category 2), the strategies that helped (Category 3), and messages for future pandemics (Category 4).

## 3. Results

### 3.1. Quantitative Analysis

As regards the problems that had arisen, 87.7% of students confirmed that the pandemic had negative impacts on their level of sedentary behavior; 58.3% referred to poor nutrition, 57.4% quantity and 69.6% quality of sleep; 25% substance abuse; 74.8% depression; 84.5% anxiety; 76.2% worries; 39% online gaming; 90% excessive screen use and 61.8% family conflicts.

The analysis of gender differences in the problems that emerged with the COVID-19 pandemic, revealed statistically significant differences in anxiety, X^2^(1, 976) = 44.979, *p* ≤ 0.001; worries, X^2^(1, 976) = 34.865, *p* ≤ 0.001; online gaming, X^2^(1, 976) = 16.685, *p* ≤ 0.001; and family conflicts, X^2^(1, 976) = 25.224, *p* ≤ 0.001. These problems were predominant in the female gender, except the online gaming, in which the male gender stood out (See Table 1).

No statistically significant differences were observed for gender in sedentary behavior, poor nutrition, sleep quantity and quality, substance use, depression, and screen overuse.

The analysis of what helped the university students to deal with the COVID-19 pandemic showed that 75.5% of them referred to family bonds, 39.7% referred to having more time, and 50.2% noted feeling less fatigue. No statistically significant differences were found in family bonds, more time or less fatigue, in the gender comparisons (Table 2).

Finally, the results on how easy/difficult was it to cope with daily routines and to balance work/study and leisure time, 51.7% of the students considered it difficult or very difficult to study/work, 43.8% for leisure, 33.8% for family interaction, and 66.1% for social interaction (See Table 3).

Statistically significant differences were found for gender, in leisure, X^2^(4, 950) = 19.931, *p* = ≤ 0.001; family interaction, X^2^(4, 951) = 13.378, *p* = ≤ 0.01; and social interaction, X^2^(4, 947) = 9.946, *p* ≤ 0.05. In leisure and family interaction, female participants showed more difficulties, in contrast to the easiness reported by the male participants. Regarding social interaction, both genders agreed that it had become difficult or very difficult. At work, no statistically significant differences were observed. Table 4 shows the detailed data.

### 3.2. Qualitative Analysis

With regard to the open-ended questions on how COVID-19 affected the well-being of first-year university students, what problems appeared, what helped them, and which were their messages for future pandemics, the qualitative data analysis performed revealed that the answers were homogeneous.

When asked about how the COVID-19 pandemic affected their well-being, first-year university students’ answers were congruent, and highlighted negative impacts, mainly on the mental, physical (inactivity), social (isolation, inability to meet new colleagues and to be with friends), and academic (demotivation, fatigue due to overwork) levels. Female participants stressed the psychological consequences of the pandemic, identifying symptoms of sadness, anxiety, depression, and irritability, as well as an increase in family conflicts. In the written statements, the following ideas stood out:
“*In physical terms by moving less, and in psychological terms by talking less to people and going out less. It also seriously affected, due to these and other factors, my school performance.*” (Female)
“*… When I spend all day in college wearing a mask, I get much more tired and have frequent migraines… When we are in a confinement, being with the family 24 h a day without being with anyone else creates more conflicts and is not healthy. Just being at home and always with the same routine and not being able to do the same as before messes with the psychological and increases the feeling of sadness.*” (Female)
“*It affected everyday life, changing routines, increasing isolation and creating a sense of constant tension in various life domains (work, family relationship, learning, and physical well-being).*” (Male)
“*I’m sick of being at home, and I’m starting to lose patience more easily with my family. I also feel more bored, which leads to other emotions restlessness, impatience, and sadness.*” (Male)

Concerning the problems that may have appeared, the number of answers was small, thus not allowing a gender analysis or data exploration. Excessive fatigue and psychological stress due to the workload were the main problems mentioned with regard to this question.

When asked about what helped to cope with the pandemic situation, young people agreed on the need to adopt coping strategies, to be able to hold out until the return to normality, stressing the importance of enjoying freedom and leisure activities, the practice of physical activity, the contact with friends and family, and the support from the university (e.g., return to face-to-face teaching, reduction in learning workload, health promotion activities, socialization). In addition, female students added the need for psychological support. Their written answers about these issues are illustrated below:
“*Contact with health professionals such as psychologists, therapists, and physicians.*” (Female)
“*Connecting (online) with friends, trying to keep social life as much as possible.*” (Female)
“*More comprehension from higher education institutions that we also need free time for ourselves. It’s not because we are more time at home that we should have more work. I need time to do nothing. Just being.*” (Male)
“*Maintaining and creating new goals, spending time with family, maintaining communication with friends and colleagues, enjoying the free time and taking time for responsibilities.*” (Male)

Certain that the situation will last for a long time, young people stressed the importance to seize life and the opportunity to become a better person, complying with the norms established by the World Health Organization, learning with the present, and hoping not to experience another pandemic again. If on the one hand, the female participants expressed greater hope for a return to normality, once again stressing the importance of psychological support in the short term, the male gender highlighted the importance of the Government’s role in fighting the pandemic, in adopting effective and coherent strategies, and of the media, in providing the population with true information. Their messages for a better future are illustrated below:
“*… Remember that we all have bad days. The important thing is to never give up and fight every day to be happy. And don’t be afraid to ask for professional help!*” (Female)
“*Maintain hygiene habits always and build healthy relationships with others, keep in touch with everyone (via online), don’t isolate ourselves.*” (Female)
“*… We’ve all trained our patience, solidarity, and versatility, with this pandemic. I think that many people took the opportunity to think a bit about how their lives were going and reordered their priorities. I hope that many people have changed for the better…*” (Male)
“*Remembering the mistakes and what we have learned with this pandemic to minimize the impact of the next one(s).*” (Male)

## 4. Discussion

The findings of this study are in line with other studies, especially in the negative impacts on the psychological level, with an increase in depressive and anxious symptomatology and worries [20,21,22]; physical level, with the adoption of sedentary behaviors (e.g., [10,35]); and excessive screen time [26,36]. Other negative effects reported by students included poor nutrition, worse sleep quantity and quality, increased family conflicts; and in a smaller proportion, substance use and online gaming. These impacts can be easily understood considering that the confinements had repercussions not only on work but also on personal life [37], forcing a reorganization of daily life, and compelling the whole family to deal with stress [38], which can affect behavioral mechanisms, leading to changes in the patterns of eating, physical activity and sleep [27].

Considering well-being models, such as Ross et al. [39]’s, which encompass the domains of (i) good health and optimal nutrition; (ii) connectedness, positive values, and contribution to society; (iii) safety and supportive environment; (iv) learning, competence, education, tools, and employability; (v) agency and resilience, we easily understand that the COVID-19 pandemic and its consequences contributed to a decrease in well-being and life satisfaction among this population [40].

In gender analysis, our findings revealed differences in anxiety, worries, online gaming, and family conflicts, with female participants showing higher scores in all these variables except for online gaming. It was also the female students that stand out in verbalizing depressive symptoms, greater sadness, and irritability. In a study by Wang et al. [41], the female gender was also associated with greater psychological symptoms. The consequences at the social level were expressed by both female and male participants, and attributed to isolation and to the inability to be with friends. Orben et al. [42] argued that adolescents and young adults may be the most affected by the social deprivation of interaction with peers, an essential component in their development. Although females were more prone to a higher risk of experiencing mental health problems, there were no statistically significant differences in the gender comparison in depression, as well as in sleep, sedentary behavior, excessive screen time, substance use and nutrition. 

Additionally, the repercussions in the academic sphere were highlighted [43], with an increase in the feelings of demotivation and fatigue associated with an excessive amount of tasks. The closure of educational establishments and the online learning methodologies adopted during the periods of confinement were associated with less time devoted to learning, greater stress, changes in relationship patterns, and lower motivation [43].

Regarding the ease in coping with the study/work routines, university students revealed greater difficulty in the study/work and social areas and less difficulty in leisure and interaction with the family. It was the female participants that revealed greater difficulty in family, social and leisure dynamics, which may explain their predominance in expressing anxious and depressive symptomatology, given that social support and coping strategies were compromised. No gender differences were found with regard to study/work.

Family connections are assumed as an important coping strategy to deal with the impacts of the pandemic on both genders, although there are no statically significant gender differences (as well as in greater time or less fatigue). The need to enjoy freedom and leisure activities, practice physical activity and maintain contact with friends was also highlighted, in line with previous studies conducted in Portugal [9,13,14]. As regards the university, students pointed out that they needed support in the reduction in the workload, in socialization to resist until the return to normality and in the readoption of the face-to-face learning methodology. The female students also stressed the necessity of psychological support at faculties.

Concerning the messages for the future, our findings suggest that the messages of young people during the first wave [9] were not fully listened to or were not given adequate responses and that most problems remain unsolved. Still unsure about the return to normality, the university students that participated in the present study stressed the importance of enjoying life and the opportunity to become better people and complying with the standards and rules of the World Health Organization. Females reaffirmed the need for mental health support, even though the provision of free psychological support was the measure most implemented by higher education institutions at a national level during the first wave of the pandemic [44,45]; on the other hand, men addressed the need for the adoption of effective and coherent strategies by the Government, and for the media to convey reliable messages.

Looking for a return to new normality (anticipated in the new 2022/2023 academic year), which is intended to be more positive and healthier, there is an urgent need to maintain/adopt institutional measures, strategies, and policies that minimize the consequences on the well-being and physical and psychological health of university students; studies that investigate the new profile and needs of post-pandemic university students; and concerted national strategies and policies to solve the problems created or highlighted by COVID-19 in this population.

## 5. Conclusions

In conclusion, this study highlighted the increase in psychological symptoms (anxiety, depression, and worries) and excessive screen time. The female students expressed symptoms of anxiety and worries, as well as greater difficulty in family relationships, even though they were more positive compared with the opposite gender at the pandemic’s end.

After the first wave of the pandemic in which problems at the academic and psychological levels were identified [9], little change was portrayed by this study of the second wave. Online classes, difficulties in socialization and high workloads are still pointed out by university students as major problems. 

In addition, psychological support remains insufficient, as the resources of the National Health Service and support created in higher education institutions [44,45] have not always been able to meet students’ needs, such as psychological support. 

For future pandemics (or any major global change), a concerted effort between the Government and the media is crucial in transmitting messages to the population, facilitating their information, understanding, and application of protective behaviors; it may also contribute to health promotion and the reduction in psychological symptoms.

## 6. Strengths and Limitations

Concerning the limitations of the study, as it was conducted during a pandemic period, its methodology was exclusively online, not allowing a thorough exploration of the responses or the registration of non-verbal information. A second limitation regards the predominance of the female gender in the sample, an increasingly normal fact in the psychosocial literature. This is something that we think may constitute a bias in the results, which benefited from the qualitative component. Another limitation concerns the population being excluded from one university (even though from different faculties), which makes a comparison with other universities in the country unfeasible.

However, we highlight also several strengths of our study such as the mixed methodology used, as a way to further understand the university students’ perspectives; the possibility to compare these study findings with findings from other studies with the same population, conducted at an international level (e.g., [20,21,22,35,46]), and the use of the computer-assisted analysis of qualitative data in the search for more transparent [47] and reliable results [48].

## Figures and Tables

**Table 1 ijerph-19-13454-t001:** Gender differences in problems that could emerge with the pandemic COVID-19.

What Problems Can Arise?	F (%) *N* = 678		M (%) *N* = 298		*N*	*χ* ^2^	*df*	*p*
	Yes	No	Yes	No				
Sedentary behavior	86.9	13.1	89.6	10.4	976	0.001	1	n.s.
Poor nutrition	58.3	41.7	58.4	41.6	976	0.001	1	n.s.
Quantity of sleep	59.1	40.9	53.4	46.6	976	2.837	1	n.s.
Quality of sleep	71.2	28.8	65.8	34.2	976	2.923	1	n.s.
Substance abuse	24.5	75.5	26.2	73.8	976	0.316	1	n.s.
Depression	74.2	25.8	76.2	23.8	976	0.433	1	n.s.
Anxiety	**89.7**	10.3	72.8	27.2	976	44.979	1	**<0.001**
Worries	**81.6**	18.4	64.1	35.9	976	34.865	1	**<0.001**
Online gaming	34.8	65.2	48.7	**51.3**	976	16.685	1	**<0.001**
Excessive screen usage	91	9	87.6	12.4	976	2.679	1	n.s.
Family conflicts	**67**	33	50	50	976	25.224	1	**<0.001**

Notes: bold = statistically significant differences and higher percentages of “Yes”; n.s. = not significant.

**Table 2 ijerph-19-13454-t002:** Gender differences in coping strategies.

What Can Help to Deal with This Situation?	F (%) *N* = 590		M (%) *N* = 268		*N*	*χ* ^2^	*df*	*p*
	Yes	No	Yes	No				
Family bonds	75.9	24.1	74.6	25.4	858	0.170	1	n.s.
More time	39.5	60.5	40.3	59.7	858	0.050	1	n.s.
Less fatigue	50.8	49.2	48.9	51.1	858	0.285	1	n.s.

Note: n.s. = not significant.

**Table 3 ijerph-19-13454-t003:** Ease/difficulty to cope with daily routines.

How Easy/Difficult Is It to Cope with Daily Routines and Work/Study/Leisure Time Balance?	1	2	3	4	5
For study/work	3	16.1	29.3	**39.4**	12.3
For leisure	7.8	24.7	23.7	**33.1**	10.7
For family interaction	5.6	**31.8**	28.9	24.1	9.7
For social interaction	2.7	13.5	17.6	**36.5**	29.6

Notes: 1 = Very easy, 2 = Easy; 3 = Neither easy nor difficult, 4 = Difficult, 5 = Very difficult; bold = statistically significant differences and higher percentages.

**Table 4 ijerph-19-13454-t004:** Ease/difficulty to cope with daily routines by gender.

How Easy/Difficult Is It to Cope with Daily Routines and Work/Study/Leisure Time Balance?	F (%) *N* = 656					M (%) *N* = 294					*N*	*χ* ^2^	*df*	*p*
	1	2	3	4	5	1	2	3	4	5				
For study/work	2.3	15.6	29.5	38.7	13.9	4.5	17.1	28.8	40.8	8.9	940	7.679	4	n.s.
For leisure	5.8	23.6	23	**35.4**	**12.2**	12.2	**27.2**	**25.2**	27.9	7.5	950	19.931	4	**≤0.001**
For family interaction	5.5	29.7	28	**25**	**11.7**	5.8	**36.3**	**30.8**	22	5.1	951	13.378	4	**≤0.01**
For social interaction	2.5	12.3	16	**37.9**	**31.4**	3.4	16.3	21.4	**33.6**	**25.4**	947	9.946	4	**<0.05**

Notes: 1 = Very easy, 2 = Easy; 3 = Neither easy nor difficult, 4 = Difficult, 5 = Very difficult; bold = higher percentages; n.s. = not significant.

## Data Availability

Not applicable.

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
