# Peer review of "COVID-19 and Lockdown, as Lived and Felt by University Students"

_ijerph, 2022, doi:10.3390/ijerph192013454_

Round 1
Reviewer 1 Report
The authors of the manuscript address a problem often explored in research. The text disseminates the experience of students at one university and promotes good practices in the field of Covid-19 and Lockdown. I have several objections to the following parts of the manuscript.
INTRODUCTION
The researchers duplicate parts of the content:
Page: 79-81 , "In an unprecedented scenario with still unpredictable medium and long-term effects, 79 namely in the development processes, this study aimed to understand young people's 80 perspectives about how the pandemic affected their group, the problems that could 81 emerge, their support and coping strategies, and messages for future pandemics"
and
page 126-129 , , ''In an unprecedented scenario with still unpredictable medium and long-term effects, 126 namely in the development processes, this study aimed to understand young people's 127 perspectives about how the pandemic affected their group, the problems that could 128 emerge, their support and coping strategies, and messages for future pandemics. ''
Page: 62-66 ,,A study focusing on the effects of the pandemic on young people's lifestyles, found 62 a decrease in physical activity, an increase in sedentary behaviors, and longer screen time 63 [21]. These findings may be associated with the higher levels of psychological distress and 64 stress experienced, which affected behavioral mechanisms, and may have lead to changes 65 in sleep patterns, eating, and practice of physical activity [22].”
and
page: 110 – 112 ,, A study focusing on the effects of the pandemic on young people's lifestyles, found 109 a decrease in physical activity, an increase in sedentary behaviors, and longer screen time 110 [21]. These findings may be associated with the higher levels of psychological distress and 111 stress experienced, which affected behavioral mechanisms, and may have led to changes 112 in sleep patterns, eating, and practice of physical activity [22].”
Please edit these duplication of the text
SUMMARY
This part of the article needs to be completed. This is the most important part of the research exploration. The conclusions are too general. It is worthwhile to organize and detail them
e.g. pp. 335-336 , "In addition, psychological support remains insufficient" Why? Please explain to what extent.
Author Response
Reviewer 1
The authors of the manuscript address a problem often explored in research. The text disseminates the experience of students at one university and promotes good practices in the field of Covid-19 and Lockdown. I have several objections to the following parts of the manuscript.
Dear Reviewer 1,
Many thanks for your time and feedback, which was of great enrichment for this work. We have added our comments below yours.
- Introduction - The researchers duplicate parts of the content:
- Page: 79-81, "In an unprecedented scenario with still unpredictable medium and long-term effects,
- 79 namely in the development processes, this study aimed to understand young people's 80 perspectives about how the pandemic affected their group, the problems that could 81 emerge, their support and coping strategies, and messages for future pandemics"
- page 126-129, ''In an unprecedented scenario with still unpredictable medium and long-term effects, 126 namely in the development processes, this study aimed to understand young people's 127 perspectives about how the pandemic affected their group, the problems that could 128 emerge, their support and coping strategies, and messages for future pandemics. ''
- Page: 62-66, A study focusing on the effects of the pandemic on young people's lifestyles, found 62 a decrease in physical activity, an increase in sedentary behaviors, and longer screen time 63 [21]. These findings may be associated with the higher levels of psychological distress and 64 stress experienced, which affected behavioral mechanisms, and may have lead to changes 65 in sleep patterns, eating, and practice of physical activity [22].
- page: 110 – 112 ,, A study focusing on the effects of the pandemic on young people's lifestyles, found 109 a decrease in physical activity, an increase in sedentary behaviors, and longer screen time 110 [21]. These findings may be associated with the higher levels of psychological distress and 111 stress experienced, which affected behavioral mechanisms, and may have led to changes 112 in sleep patterns, eating, and practice of physical activity [22].”
Please edit these duplication of the text
We really sorry for the oversight, was a mistake transporting word document to template. Duplicate content has been removed.
- Summary: This part of the article needs to be completed. This is the most important part of the research exploration. The conclusions are too general. It is worthwhile to organize and detail them.
The aim, results and conclusions have been improved - lines 28 to 50.
- g. pp. 335-336, "In addition, psychological support remains insufficient" Why? Please explain to what extent.
Information complemented - Lines 371-373 - In addition, psychological support remains insufficient, being that the resources of the National Health Service and support created in higher education institutions [44-45] have not always been able to meet students' needs, like psychological support.
Many thanks. Best
Reviewer 2 Report
I would like to thank the authors for this research that aimed to understand young people’s perspectives about how the pandemic affected their group, the problems that could emerge, their support and coping strategies, and messages for future pandemics.
The research subject is timely, innovative, and highly interesting. It also fits the aim and scope of the journal.
The research is well designed and follows a sound scientific research method. However, several adjustments are needed.
The abstract needs several clarifications.
You said: “this study aimed to understand young people’s perspectives about how the pandemic affected their group, the problems that could emerge, their support and coping strategies, and messages for future pandemics”. You need to simplify the aim of the research.
Design and participants: The whole paragraph is cited for the second time. Repetition. Data of this section is missing. Is it a mistake?
The whole manuscript needs professional language editing. Several errors were detected.
Conclusion: Write a structured paragraph not under bullets.
Other minor comments are directly attached to the manuscript.

Author Response
Reviewer 2
I would like to thank the authors for this research that aimed to understand young people’s perspectives about how the pandemic affected their group, the problems that could emerge, their support and coping strategies, and messages for future pandemics.
The research subject is timely, innovative, and highly interesting. It also fits the aim and scope of the journal.
The research is well designed and follows a sound scientific research method. However, several adjustments are needed.
Dear Reviewer 2,
Many thanks for your time and feedback, which was of great enrichment for this work. We have added our comments below yours.
- The abstract needs several clarifications.
You said: “this study aimed to understand young people’s perspectives about how the pandemic affected their group, the problems that could emerge, their support and coping strategies, and messages for future pandemics”. You need to simplify the aim of the research.
The objective has been reformulated. - Lines 28 and 29 - The present work aims to explore gender differences in how first-year college students experienced and felt Covid-19 and the second confinement. Results and conclusions were also improved.
- Design and participants: The whole paragraph is cited for the second time. Repetition. Data of this section is missing. Is it a mistake?
We really sorry for the oversight, was a mistake transporting word document to template. Duplicate content has been removed.
The whole manuscript needs professional language editing. Several errors were detected. Conclusion: Write a structured paragraph not under bullets.
English has been revised.
Other minor comments are directly attached to the manuscript.
Month and year were added after second confinement – lines 26 and 27
The aim were reformulated - The present work aims to explore gender differences in how first-year college students experienced and felt Covid-19 and the second confinement.
The repetition in design and participants was a mistake, I’m sorry.
The sentence “mainly negative impacts at the psychological level” has been reformulated – lines 279 to 283 - Findings on how the pandemic affected first-year university students, the prob-lems that arised, their support and coping strategies, and the messages for future pan-demics, revealed, of this study are in line with other studies, mainly especially in the negative impacts onat the psychological level, with an increase in depressive and anx-ious symptomatology and worries [20-22]
According to your suggestions, the following references were integrated:
- Kooli, C. COVID-19: Public Health Issues and Ethical Dilemmas. Ethics Med. Public Health 2021, 17 (100635), 100635. https://doi.org/10.1016/j.jemep.2021.100635.
- Maqsood, A.; Saleem, J.; Butt, M. S.; Shahzad, R. B.; Zubair, R.; Ishaq, M. Effects of the COVID-19 Pandemic on Perceived Stress Levels of Employees Working in Private Organizations during Lockdown. Avicenna2021, 2021 (2). https://doi.org/10.5339/avi.2021.7.
- Kooli, C. Challenges of Working from Home during the COVID-19 Pandemic for Women in the UAE. Public Aff.2022, e2829. https://doi.org/10.1002/pa.2829.
Conclusions are now without bullets.
Reviewer 3 Report
Introduction
: The authors should provide more information on the association between COVID-19 and the prevalence of negative mental health in university students.
1) Restricted to their homes, the consequences at physical and psychological, ~ academic levels soon appeared [7-13].
: Giving detailed examples can help the reader understand.
2) ~health and well-being of this group, ~ restrictions on social contacts~
: The subject of this study is university students. Authors should describe the content relevant to the research subject.
Materials and Methods
: Various results were presented in this study, but little explanation was found on the data collection method. A detailed description of the various tools described in the results is needed to improve the validity of the study.
1) 2.1. Design and Participants
Declared a pandemic disease on 11 March 2020 [1], ~ and messages for future pandemics.
: It is recommended to delete the part already mentioned in the introduction. Duplicate content is boring.
2) 2.2. Measures
: How was the online questionnaire collected? Add a description of the details (scale, form, etc.).
Discussion
There is a lack of discussion compared to the reported results. Authors should add (even if not statistically significant) a discussion of the various outcomes.
Author Response
Reviewer 3
Dear Reviewer 3,
Many thanks for your time and feedback, which was of great enrichment for this work. We have added our comments below yours.
- Introduction - The authors should provide more information on the association between COVID-19 and the prevalence of negative mental health in university students.
The following references were added:
- Kooli, C. COVID-19: Public Health Issues and Ethical Dilemmas. Ethics Med. Public Health 2021, 17 (100635), 100635. https://doi.org/10.1016/j.jemep.2021.100635.
- Marques, G.; Drissi, N.; Díez, I. de la T.; de Abajo, B. S.; Ouhbi, S. Impact of COVID-19 on the Psychological Health of University Students in Spain and Their Attitudes toward Mobile Mental Health Solutions. J. Med. Inform.2021, 147 (104369), 104369. https://doi.org/10.1016/j.ijmedinf.2020.104369.
- Marahwa, P.; Makota, P.; Chikomo, D. T.; Chakanyuka, T.; Ruvai, T.; Osafo, K. S.; Huang, T.; Chen, L. The Psychological Impact of COVID-19 on University Students in China and Africa. PLoS One2022, 17 (8), e0270824. https://doi.org/10.1371/journal.pone.0270824.
- Li, Y.; Wang, A.; Wu, Y.; Han, N.; Huang, H. Impact of the COVID-19 Pandemic on the Mental Health of College Students: A Systematic Review and Meta-Analysis. Psychol.2021, 12, 669119. https://doi.org/10.3389/fpsyg.2021.669119.
- Branquinho, C.; Santos, A. C.; Noronha, C.; Ramiro, L.; de Matos, M. G. COVID-19 Pandemic and the Second Lockdown: The 3rd Wave of the Disease through the Voice of Youth. Child Indic. Res.2022, 15 (1), 199–216. https://doi.org/10.1007/s12187-021-09865-6.
- Restricted to their homes, the consequences at physical and psychological, ~ academic levels soon appeared [7-13]. Giving detailed examples can help the reader understand.
The sentence were complemented – lines 58 to 61 - Restricted to their homes, with mandatory curfews and closure of all non-essential services, and with few resources from the public health system (a global scenario) [7], the physical and psychological, social, family, and academic consequences rapidly emerged in students’ lives [8-14].
- health and well-being of this group, ~ restrictions on social contacts~
In the sentence reformulation, that part was deleted. – lines 49 to 57
Declared a pandemic disease on 11 March 2020 [1], the SARS-Cov-2 coronavirus has forced the implementation of numerous measures to mitigate the virus spread [2]. In Portugal, the first emergency state was declared on 18 March 2020 [3], as a result of a calamity situation, prompting mandatory confinement of patients or those infected with SARS-CoV-2; isolation of risk groups; and restriction of circulation on public roads to other citizens (who moved to a telework regime, whenever their duties al-lowed). Students, who were unable to pursue face-to-face education with the closure of educational institutions, experimented an online methodology, adopted by numerous countries [4-5]. Its impacts and methodology continue to be the focus of studies [6].
- The subject of this study is university students. Authors should describe the content relevant to the research subject.
- Materials and methods: Various results were presented in this study, but little explanation was found on the data collection method. A detailed description of the various tools described in the results is needed to improve the validity of the study.
Was a mistake, and some data wasn´t in this work, I’m sorry. Design and participants lines 96 to 117 and measures – lines 119 to 132 are now corrected.
- Design and Participants - Declared a pandemic disease on 11 March 2020 [1], ~ and messages for future pandemics. It is recommended to delete the part already mentioned in the introduction. Duplicate content is boring.
Was a mistake, and that data wasn´t in this work, I’m sorry. This section is now suitable in lines 96 to 116.
- Measures - How was the online questionnaire collected? Add a description of the details (scale, form, etc.).
Lines 118 to 120 - An online survey conducted through the Google Forms platform was designed based on the following questionnaires: Nutrition [31], Health Behaviour in School-aged Children/World Health Organization [32-33], and Health Literacy [34].
- There is a lack of discussion compared to the reported results. Authors should add (even if not statistically significant) a discussion of the various outcomes.
All the outcomes are now included in the discussion.
Round 2
Reviewer 1 Report
The latest version of article (dated September 2022), includes amendments, taking into account some of the remarks and postulates indicated.
Regards
Reviewer 2 Report
I would like to thank the author for making the suggested changes.
Reviewer 3 Report
This study is a much needed study on mental health problems during the pandemic.